# Therapeutic Strategies of Primary Molar Infraocclusion: A Systematic Review

**DOI:** 10.3390/children10030582

**Published:** 2023-03-18

**Authors:** Assunta Patano, Angelo Michele Inchingolo, Claudia Laudadio, Daniela Azzollini, Grazia Marinelli, Sabino Ceci, Giulia Latini, Biagio Rapone, Alessio Danilo Inchingolo, Antonio Mancini, Francesco Inchingolo, Daniela Di Venere, Gianluca Martino Tartaglia, Gianna Dipalma, Giuseppina Malcangi

**Affiliations:** 1Department of Interdisciplinary Medicine, University of Bari “Aldo Moro”, 70124 Bari, Italyangeloinchingolo@gmail.com (A.M.I.);; 2Department of Biomedical, Surgical and Dental Sciences, School of Dentistry, University of Milan, 20100 Milan, Italy; 3UOC Maxillo-Facial Surgery and Dentistry, Fondazione IRCCS Cà Granda, Ospedale Maggiore Policlinico, 20100 Milan, Italy

**Keywords:** paediatric dentistry, public health dentistry, orthodontics, tooth ankylosis, tooth, unerupted, tooth, impacted, tooth abnormalities

## Abstract

Background: Infraocclusion of deciduous molars is a clinical disturbance that occurs during primary and mixed dentition and has some orthodontic implications. Infraoccluded teeth are believed to be potential sites of malocclusion, with a risk of tipping neighbouring teeth and losing space. This systematic review aims to analyse the management of primary molars infraocclusion and to provide updated guidelines. Methods: A literature search was performed using PubMed, Scopus, and Web of Science databases from 1 January 2017 to 28 November 2022. The inclusion criteria were: studies only on human subjects, open access studies, case reports, randomised trials, retrospective, observational studies, and English language. Results: A total of 372 publications were identified from the databases and a final number of nine studies were included in the review for qualitative analysis. Conclusion: Management of patients suffering from infraocclusion depends on the severity, age at diagnosis, and presence of succeeded premolars. Early diagnosis of infraoccluded primary elements is fundamental and cannot be postponed. Preservation of the primary molars may be a valid option with long-term stability if there is no or moderate primary molar infraocclusion, root resorption of less than half of the root, and no decay or restoration.

## 1. Introduction

Infraocclusion is a clinical condition that occurs during mixed dentition in which a primary molar appears inferior to the occlusal plane without apparent physical obstacles [1]. The infraocclusion process can be related to a disturbance of the tooth’s eruption or occurs after the full eruption of the teeth. Many terms are used in literature to describe this condition; the most common are ‘submerged tooth’, ‘retentive tooth’, and ‘ankylosed tooth’ [2,3,4,5].

### 1.1. Incidence and Etiology

The prevalence of infraocclusion has been variously reported as ranging between 1.3% and 8.9% in growing patients, with no gender predilection noted [6,7]. It is commonly observed during the developing dentition, and the peak prevalence is at the age of eight-to-nine years old [8]. Mandibular molars are affected more than maxillary molars, and infraocclusion often presents bilaterally [9]. Some studies focused on the link between the tooth involved and the age of the child at diagnosis: The mandibular first primary molar is the most prevalent tooth before nine years of the patient’s age; the mandibular second primary molar is the most commonly affected when the patient is over nine years old [10].

Not rare is the association between infraocclusion and other dental anomalies, in particular, the absence of permanent successor teeth (65.7%) [11,12,13,14]. Other anomalies reported in the literature are the ectopic eruption of first permanent molars, peg laterals, enamel hypoplasia, and palatal displacement of maxillary canine [15,16]. The occurrence of infraocclusion among siblings suggested genetic implications [17].

Infraocclusion is widely believed to be connected to ankylosis [5,18,19,20]. There is no evident aetiology of infraocclusions in the literature; however, it is possible to identify a specific histological profile. Histologic studies on primary molars in infraocclusion revealed degenerative changes in the pulp, such as fibrosis, calcifications, and areas of reduced cell activity in the periodontal ligament [21,22]. Ankylosis is histologically defined as the fusion of cementum/dentin to the bone in at least one area, resulting in the loss of the periodontal ligament space. Damage to the Hertwing epithelial root sheath may occur during the eruption process, causing direct contact of cementum with bone, then ankylosis, and, consequently, a vertical stagnation of tooth eruption resulting in an infraocclusion of the tooth [23,24,25]. Overall, ankylosis convoys primary molars in infraocclusion with an incidence between 1.3 to 3.8% [13].

### 1.2. Clinical Features and Diagnosis

Detection of infraocclusion is generally established through the clinical finding of the tooth submersion under the clinical plane, percussion testing, and radiographic evidence. When percussed with metallic instruments, the typical sound of a “cracked teacup” characterised the infraoccluded tooth [26]. Radiographic parameters are usually obtained from orthopantomography and periapical radiographs. Radiographic criteria in orthopantomography are: the step between the infraoccluded tooth and the adjacents, the common bilateral presentation, and the presence of obliteration of the periodontal ligament [27,28]. Examination of an orthopantomography can help identify other dental anomalies. Cone-beam computed tomography (CBCT) images can be used as an additional diagnostic tool to best visualise the impacted tooth before surgical procedures.

The severity of infraocclusion is classified as slight, moderate, or severe according to the relationship of the occlusal surface of the tooth relative to adjacent teeth [19] (Figure 1). When the affected tooth is located 1 mm below the expected occlusal plane, the infraocclusion is considered mild; when it levels with the interproximal contact point of adjacent teeth, the defect is assessed as moderate; infraocclusion is described as severe when the occlusal surface is placed with or below the interproximal gingiva tissue of other teeth [29,30].

### 1.3. Orthodontic Implications

If infraocclusions are left untreated, several repercussions on the occlusion could be expected, and successful management may involve invasive procedures [31,32] (Figure 2). Dental agenesis, microdenture of the upper lateral incisors, the palatal position of the upper canines, and distal inclination of the lower second premolars can all be caused by an infraoccluded deciduous tooth [33,34]. The development of a lateral open bite due to vertical growth inhibition of the alveolar process is the most frequent orthodontic malocclusion related to infraocclusion [35,36]. Clinical disturbances may include the delayed eruption of successor with ectopic displacement, caries and periodontal diseases, possible resorption of proximal root surfaces, incomplete alveolar process development, and lack of normal mesial drift. The infraoccluded tooth appears depressed with the tipping of the adjacent teeth and the overeruption of opposing teeth [33,37,38]. This causes a reduction in arch length and loss of space, especially in cases of severe infraocclusion of deciduous second molars in mixed dentition [33,39]. A significant deviation of the dental inter-incisor midline toward the affected side when the affected tooth has been left in place is another consequence reported in the literature [40].

The progression with age must be considered, monitoring and recording the progression, and amount rate in every appointment: An infraoccluded tooth might become ankylosed, not responding to orthodontic forces and making intervention more traumatic [41,42].

This research aims to investigate and clarify the recent evidence regarding primary molar infraocclusion in its management and resolution to give an updated guideline.

## 2. Materials and Methods

The present systematic review was performed according to the principles of the PRISMA [43] and the International Prospective Register of Systematic Review Registry guidelines (PROSPERO ID n°400782). PubMed, Scopus, and Web of Science were searched to find papers matching our topic dating from 1 January 2017 to 28 November 2022, with English-language restriction. The search strategy was developed using a combination of words that fit the purpose of our investigation, whose primary focus was on the therapeutic strategy of infraocclusion of a primary molar. Hence, the following Boolean keywords were used: “infraocclus *” OR “ankylos *” OR “submer *” OR “secondary retention” AND “molar”(Table 1).

The authors checked the titles and complete texts of any papers that might be relevant. All acceptable studies were examined by two independent reviewers (A.P., D.A.), working in duplicate, using the following inclusion criteria: (1) human subjects studies; (2) open-access studies that are available to all researchers without a subscription; (3) case reports, randomised trials, retrospective studies, and observational studies; and (4) research that examined the relationship between primary molar infraocclusion and its treatment. Reviews, letters to the editors, research that considers permanent teeth, studies that investigate the infraocclusion diagnosis process, and prevalence or correlation studies were all excluded. Disagreements between the investigators regarding the articles’ selection were adequately discussed and resolved by adjusting the inclusion and exclusion criteria.

## 3. Results

### General Characteristics of the Articles Included

A total of 372 publications were identified from the following databases, including Pubmed (106), Scopus (118), and Web of Science (148), which led to 226 articles after removing duplicates (146). The title and abstract analysis excluded a total of 192 publications. The remaining 34 articles were successfully sought for retrieval and were assessed for eligibility by the authors. Twelve publications were excluded from the process because they were off-topic, eight others because they were prevalence and correlation studies, and five because they focused on the diagnosis and not on the treatment of infraocclusion. A final number of nine studies were included in the review for qualitative analysis (Figure 3).

The study data were selected by analysing the study design, sample characteristics (gender and age), infraoccluded tooth and degree of infraocclusion, the presence or absence of a successor, the type of treatment, and the outcome obtained (Table 2).

## 4. Discussion

### 4.1. Interpretation of the Results

The articles that were analysed in this systematic review had, as the type of treatment adopted, a surgical approach involving the extraction of the deciduous elements in cases where the permanent element was present. Only in cases in which agenesis of the permanent element was diagnosed was the choice made to preserve the deciduous tooth in the long term in cases deemed appropriate. This choice must be evaluated case-by-case, especially with other diseases or specific clinical conditions.

Below is an analysis and summary of the selected articles to better evaluate the therapeutic choices and outcomes. A wide use of different terms to indicate similar or overlapping conditions was found.

#### 4.1.1. Infraocclusion of Primary Molars

The underlying aetiology infraocclusions of primary molars are various. According to Kjær [44], eruption problems can occur in the following three developmental phases:**Phase 1**—migration period (from tooth bud to early root formation). Two conditions can be encountered:
○Ankylosed deciduous molars are extracted as soon as diagnosed. Sometimes, premolars tooth germs are extracted. If they are not extracted, the permanent crown can be malformed;○Not ankylosed deciduous molars are extracted when root development of the premolars has begun. Premolars should be monitored and, if feasible, saved.

In both cases, the premolar tooth bud could move to ectopic positions due to the altered development of permanent dental lamina and tooth bud.

**Phase 2** (the period before the premolar penetrates the gingiva). There were two possible scenarios:
○Abnormal premolar eruption and absence of primary molars exfoliation. Unclear are the causes behind the retained primary molars. There are only assumptions such as:
▪Segmental bone dysplasia, in which premolar eruption is delayed → primary molars should be removed once premolar eruptive movements have begun;▪Ectoderm deviations → extraction of primary molars before root closure of premolars.


In both cases, primary molar extraction was the elective treatment, observing an eruption of premolars in young subjects.


○Abnormal premolar eruption after extraction of primary molars. If, for example, there were root resorptions of the permanent root, with repair attempts or a prolonged eruption stop, the ankylosed premolars could not erupt and were therefore extracted.


**Phase 3** (premolar eruption after penetration of gingiva). Premolars ankylosed could be observed. The cause is generally attributed to resorption of the permanent periodontal membrane due to malfunction in the peripheral nerve tissue (genetically determined or caused by virus attacks) or primary failure of tooth eruption (PFE) (defect in the receptor for parathyroid hormone, PTH1R). Surgery might be the only treatment and, in very few cases, orthodontic treatment.

Therefore, abnormalities of the primary molars could affect the erupting premolar in the first two phases.

Garcovich [45] reported the case of a young boy with Class I canine occlusion, increased OVJ, and anterior crowding with a 4.6 mesially tipped in the space of 8.5. A simple orthodontic approach was performed to widen the space and extract the submerged second deciduous molar (8.5). After three months, the space was regained using a band with a buccal tube on the 4.6, a 0.014” round NiTi wire bonded to the 7.5, and extraction of 8.5 was conducted. A 0.018” Green Australian archwire was then used to maintain adequate space for the premolar eruption. The second premolar erupted a year later, and the final OPG X-ray revealed the eruption of the second premolar and uprighting of the 3.6 and 3.3. This study indicates the efficacy of light orthodontic therapy in restoring space and promoting the natural eruption process.

Atia’s article [46] reports the case of a patient who was accidentally diagnosed with a deciduous molar with severe infraocclusion and described the surgical-orthodontic approach. It also demonstrates and suggests using CBCT as an ancillary diagnostic tool. The described clinical case concerned a young patient who showed the simultaneous presence of a severe infraoccluded deciduous second molar and its corresponding erupted permanent premolar in the palatal position. Initially, an endoral X-ray was performed at the level of the right maxillary arch. This showed the presence of a radiopacity in the apical position near the almost unerupted second premolar crown. A CBCT scan was performed to identify the radiopacity, resulting in the presence of the crown of the infraoccluded primary second molar. In agreement with the parents, removing both elements was the surgical route. The deciduous molar was extracted to avoid the risk of cystic evolution or root resorption of the neighbouring permanent teeth. On the other hand, the second permanent premolar was removed due to the lack of space in the upper arch. Unfortunately, the parents decided against post-surgical orthodontic treatment. The paper’s authors encourage early intervention with the extraction of the deciduous element before it becomes infraoccluded and the placement of a space maintainer to allow the eruption of the corresponding permanent. Alternatively, in the case of ankylosed teeth, a conservative approach is also possible, with a crown build-up of the element or a steel crown placement to avoid tipping the crowns of adjacent elements and over-eruption of the antagonist. All these interventions can prevent extensive bone loss due to the removal of infraoccluded/anchylosed teeth with the associated complications. In conclusion, monitoring the correct shift of dental elements is fundamental and preventive against these eruptive anomalies and complications.

#### 4.1.2. Association between Infraocclusion of Deciduous Molars and Other Conditions

##### Ankylosis

This case study [47] shows a severe example of dental ankylosis that resulted in mandibular second premolar dislocation and necessitated space-regaining therapy before extraction. The patient’s condition was characterised by impaction of the second primary molar and second premolar. Orthodontic treatment was also performed to facilitate extraction of the deciduous molar and prevent damage to adjacent teeth. The mandibular left second premolar eruption was monitored, and fenestration and traction procedures were considered if needed. The loss of alveolar bone height due to ankylosis was expected to recover with the eruption of the permanent successor, but the mandibular left second premolar was discovered to be wholly impacted and in contact with the root of the mandibular left first permanent molar, necessitating careful monitoring as well as possible fenestration and traction procedures. The evolution of the permanent premolar has yet to be evaluated at the time of the study.

Saitoh [48] described a child with a posterior open-bite case due to the infraocclusion of an ankylosed second primary molar. Impaction or infraocclusion of upper or lower molars, or both, can produce a posterior open bite. The treatment plan included the first permanent molar uprighting and distalization to gain the proper space for the second premolar. Then the first primary molar was extracted, and a tongue block fence was added to the lingual arch. Then, thanks to a surgical gingival-alveolar bone approach, the second premolar emerged naturally five months after the excision of the affected second primary molar.

##### Hypophosphatasia

Hamada [49] reported that there may be other causes underlying the infraocclusion of primary molars. These can be submerged and ankylosed due to hypophosphatasia (HPP), a rare genetic disorder. HPP is usually characterised by early exfoliation of deciduous teeth, particularly the anterior teeth, disturbed dental cementum formation, low serum alkaline phosphatase concentration, and bone hypomineralisation. It is hypothesised that this disturbed cementum formation and altered occlusal forces could lead to susceptibility to early exfoliation of anterior teeth and ankylosis involving posterior teeth. The infraoccluded primary molar was extracted in the reported case, and the respective permanent premolar showed eruption tendency.

##### Hypodontia

Hypodontia is the congenital lack of at least one permanent tooth, excluding third molars, while agenesis refers to a single tooth [50]. A case of severe hypodontia was reported by Dr Ping Lin’s group [51]. In patients with severe dental agenesis, a type of dental anomaly in which individuals lack teeth due to developmental failure caused by genetic or environmental factors, other dental anomalies, including infraoccluded or ankylosed teeth, may co-occur. This case involved orthodontics, periodontics, and prosthetics treatments, with the periodontist playing an essential role in rehabilitating edentulous regions with implants in combination with various augmentation procedures (sinus lift and grafts) of hard and soft tissues and frenectomy. All the primary teeth were extracted, and the molars were ankylosed. However, removing retained deciduous molars without successors is only sometimes necessary, and these elements remain in good health over time. In this case, they were extracted because no restorative treatments could be performed due to wide interdental spaces. A good summary regarding what to do in cases of agenesis of the second permanent premolar and the presence of secondary primary molars was proposed by Bjerklin [37] in his expe corner article. In non-extractive cases, is it possible to keep the primary molar in situ if there is/are:No crowding with Angle Class I occlusion, normal overbite, and overjet with the primary molar in overall good condition;Vertical deep bite, no crowding, mandibular clockwise rotation, and incompatible growth pattern with extractions, no or minor primary molars infraocclusion, root resorption less than half of the root, and no caries or fillings of these elements. Preservation of the primary molars may be a good option with long-term stability;Augmented overbite, reduced lower facial height, retroclination of the lower incisors, ipodivergence, and no crowding in the lower arch.

The decision to leave a primary molar in situ (i.e., without extracting it) is recommended when extracting a primary second molar early, before the age of 12–13 years. This choice may be opted for in cases where agenesis (missing) of the lower second premolar was diagnosed late or if the patient’s parents refused extraction.

On the other hand, extractions are suggested when there are:Severe issues such as root resorption, infraocclusion, and decays of the roots of the second primary molar (at ages 10–11 years);Crowding with mesial tipping of the first mandibular molar, normal or minimal overbite, Angle Class I occlusion, regular growth pattern, and normal incisal inclination. In addition, consider extraction of the remaining three second premolars.


In extractive cases, treatment options are:


Fixed orthodontics space closure, also with TADs aid;Spontaneous mesialisation of permanent molars: Between the age of 8–9 years, before the extraction of the second deciduous molar, slicing and then hemisection of its crown is advised. In this way, there is a controlled mesialization of the first permanent molar, avoiding crown tipping, especially if the extraction is performed after the completed root development of the first permanent molar;Dental element autotransplantation when the space closing is challenging or impossible. It has been seen that the maxillary third molar is an excellent candidate for insertion at the level of the mandibular second deciduous molar;Placement of an implant and prosthetic crown. This is a choice to be made when alveolar bone development has been completed (age > 20 years) because the implant blocks the growth of the alveolar process;Fixed prosthesis: such as implants. This option interferes with alveolar bone growth.


The relationship between premolar deciduous and primary molar infraocclusion has been proven to be evident.

The work of Hvaring and Birkeland [50] aimed to assess the stability of deciduous teeth in patients affected by severe hypodontia. A large group of patients (42) who had at least one persistent deciduous tooth was evaluated at baseline and follow-up. Only root resorption, infraocclusion, and restorations were studied when assessing deciduous teeth. Furthermore, the analysis did not consider malocclusions or conditions that could have interfered with the stability of the deciduous elements. Teeth with short roots proved to be more persistent, while infraoccluded ones were more likely to be lost. Except for three teeth that were replaced by space closure, most of the replacements were dental implants. The study concludes that retention of healthy primary canines and molars is trustworthy and that early infraocclusion is unfavourable to tooth stability.

This condition was also described by Ng et al. in their work [52], where a treated patient presented severe infraocclusion of 4.5, 8.5 with its ankylosis, over-retained 7.5, and congenitally missing 3.5. The therapeutic decision was to extract the 8.5, 7.5, and the 4.5. The 4.5 would have been difficult to recover with orthodontic traction and could have been damaged by the removal of 8.5. Furthermore, the presence of the lower third molars made it possible to schedule their mesialisation and the achievement of a correct occlusion. In Ng et al.’s work, it is possible to evince that extraction and implant insertion are not always the only way forward, mainly due to the young age of patients and the presence of satisfactory treatment alternatives (e.g., orthodontics), as suggested by Bjerklin’s article.

### 4.2. Limitations of the Evidence Included in the Review

Limitations of the evidence included in this systematic review were linked to the studies considered (case reports). In addition, there were few articles regarding the treatment of infraoccluded primary elements in the last 5 years. Some studies (Nagayama et al. and Hamada) have not considered the permanent premolar’s outcome at the study’s time. Furthermore, only a few of the reported articles analysed the entire orthodontic treatment adopted after the resolution of infraocclusion.

### 4.3. Implications of the Results for Practice, Policy, and Future Research

The importance of proper timing and determination of the severity of infraocclusion is indisputable to minimise complications and prevent damage to adjacent teeth [47]. If an infraoccluded tooth is classified as slight or moderate, follow-up should be carried out. Severe infraocclusion represents a criterion for an invasive procedure such as extraction, in order to prevent orthodontic consequences.

In cases of ankylosis, careful monitoring of erupting direction is necessary to ensure proper tooth movement. Ankylosed deciduous molars typically show mild to moderate progression and can be followed conservatively, as most of these teeth eventually exfoliate spontaneously due to the eruption of the permanent successor. When an impacted deciduous tooth is detected, it is essential to evaluate its position relative to the permanent successor and to confirm the presence of ankylosis through X-ray findings. In cases of ankylosis, extraction of the impacted tooth may be necessary, but care must be taken not to damage the permanent successor’s tooth germ. Space regain therapy or orthodontic treatment may be required to facilitate extraction and minimise the risk of damaging adjacent teeth. Early intervention with the extraction of the primary element is suggested before it becomes infraoccluded. It is also recommended to place a space maintainer to allow the eruption of the corresponding permanent [46,53] or to carry out a light orthodontic therapy [45,54].

However, cases of severe ankylosis are rare and the affected tooth is unlikely to exfoliate on its own. Left untreated, severe ankylosis can result in a range of complications, including inhibition of eruption of the permanent successor, tipping of adjacent teeth, decreased dental arch length, or supereruption of an opposing tooth leading to malocclusion.

However, generally speaking, it is possible to keep an over-retained primary molar when the corresponding permanent is missing, but it must be stable, unrestored, with a good root component, and not infraoccluded. It is reported that, after the age of 20, over-retained primary molars showed minimal changes throughout the years with an excellent long-term life [52]. Mandibular primary canines and second molars were the most likely to be replaced, although maxillary canines and molars were the least probable. When dealing with patients with infraoccluded deciduous molars, a correct clinical approach would involve extracting these deciduous elements, if infraocclusion > 2–3 mm, before the growth spurt, and being optimistic about primary teeth with a short root. A primary element in proper occlusion with a short root can serve as an effective (semi) permanent replacement [50,55].

Every treatment option must be customized for the specific needs of each patient, considering diseases, and particular clinical situations. For instance, leaving the infraoccluded tooth in olygodonthia or dental agenesis as a space maintainer may be beneficial.

Future research suggestions are to evaluate the comprehensive resolution of the case as longitudinal studies.

## 5. Conclusions

The cause of infraoccluded deciduous molars is not fully understood, and the best resolution approach has yet to be determined [48];Early diagnosis of infraoccluded primary elements is fundamental and cannot be postponed [56];Preservation of the primary molars may be a valid option with long-term stability if there is no or moderate primary molar infraocclusion, root resorption of less than half of the root, and no decays or restoration [52];Good clinical situations for early extractions: class I, crowding, normal OVB, and OVJ, normal intermaxillary relation, normal incisor inclination;Primary molars preservation in case of delayed extractions after 12–13 years; diastemas; no or minor infraocclusion; no caries or filling; vertical deep bite cases; class III tendency or plane mandibular angle; root resorption < than half of the root.

The correct therapeutic strategies for primary molar infraocclusion are conservative, in case of lack of the corresponding premolar, with preservation of the deciduous element, if there is a moderate type of infraocclusion and a stable dental condition with adequately represented roots. Deciduous element extractions are performed when there is a good class ratio with crowding and normal cephalometric parameters. Placement of a space maintainer to allow the eruption of the corresponding permanent is mandatory in case of early extractions.

## Figures and Tables

**Figure 1 children-10-00582-f001:**
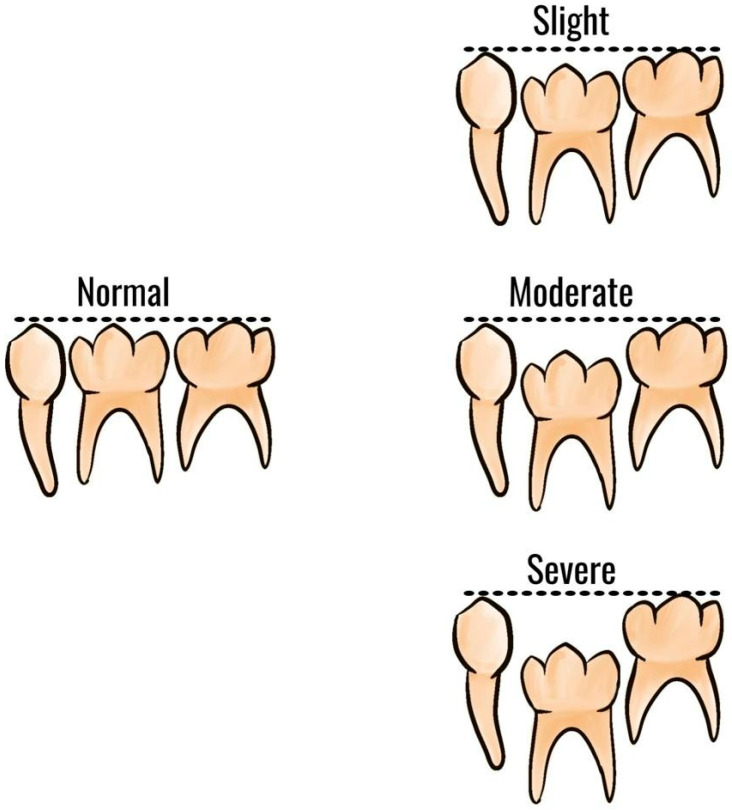
Schematic classification of infraocclusion of deciduous molars.

**Figure 2 children-10-00582-f002:**
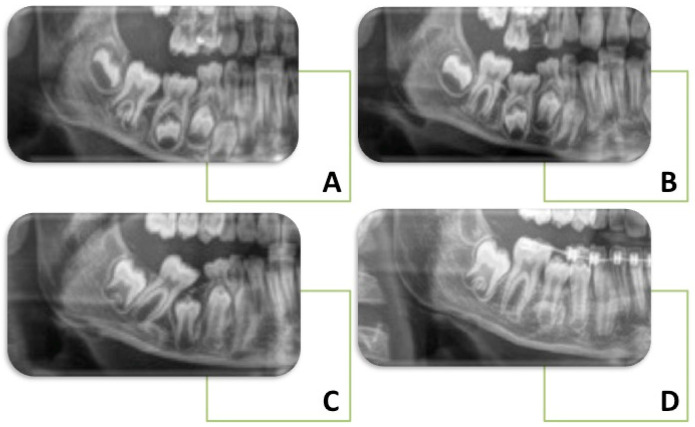
Orthodontic implications: X-ray sequence of the lower second primary molar in infraocclusion: Diagnosis of infraocclusion during the early mixed dentition (**A**); Development of lateral open bite due to the infraocclusion (**B**); Tipping of the adjacent teeth after the extraction of infraoccluded tooth (**C**); Space gained through the fixed appliance and spontaneous eruption of permanent premolar (**D**).

**Figure 3 children-10-00582-f003:**
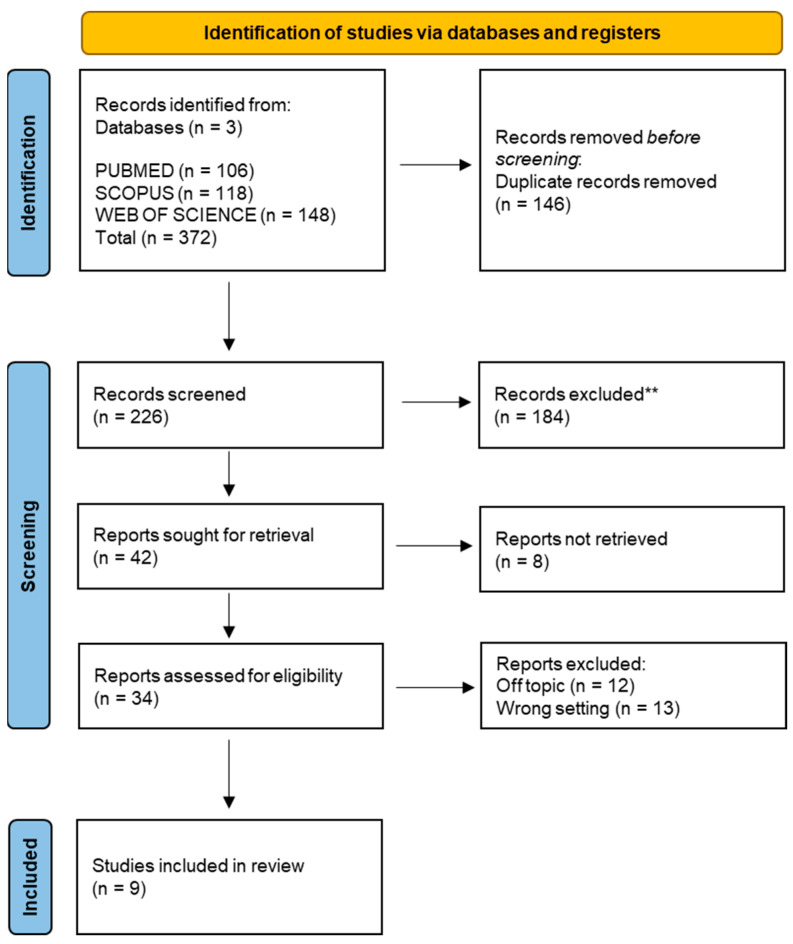
PRISMA flowchart diagram of the inclusion process. Literature search Preferred Reporting Items for Systematic Reviews and Meta-Analyses (PRISMA) flow diagram.

**Table 1 children-10-00582-t001:** Database search indicators.

Articles screening strategy	KEYWORDS: (“infraocclus *” OR “ankylos *” OR “submer *” OR “secondary retention”) AND “molar”. Timespan: from January 2017 up to November 2022. Electronic Databases: PubMed, Scopus, Web of Science.

* the asterisk is used to encompass all words beginning with the noun root.

**Table 2 children-10-00582-t002:** Study characteristics and results.

Author, Year	Type	Subject	Main Issue	Treatment	Outcome
Atia et al., 2018	C. R.	F., 12 years	Severe infraoccluded upper second deciduous molar with successor	Extraction	Resolution of dental crowding
Garcovich et al., 2019	C. R.	M., 8 years	Severe infraoccluded lower second deciduous molar with successor	Extraction and wait and watch	Premolar spontaneous eruption
Hamada et al., 2020	C. R.	M., 9 years	Mild infraoccluded lower primary molar with the successor	Extraction and wait and watch	Premolar tendency to erupt
Hvaring and Birkeland 2019	L. O. S.	24 F., 26 M., Average 13.5 years	Slight/mild infraoccluded upper/lower deciduous molar teeth without successor	No extraction	Infraoccluded teeth tend to remain over time
Kjær 2021	C. R.s	n. a.	Phase 1: Ankylosed primary molars Not ankylosed primary molars with successor	Extraction	Premolar tendency to erupt
			Phase 2: Retained primary molars with successor	Early extraction	Premolar tendency to erupt
Lin et al., 2019	C. R.	M., 18 years	Mild infraoccluded upper/lower primary molar without a successor	Extraction	Implant-prosthetic rehabilitation
Nagayama et al., 2022	C. R.	M., 10 years	Severe infraoccluded lower second deciduous molar with the successor	Extraction and wait and watch	n. a.
Ng et al., 2022	C. R.	M., 14 years	Severe infraoccluded lower second deciduous molar with successor	Extraction	Orthodontic treatment for space closure
Saitoh et al., 2017	C. R.	F., 7 years	Severe infraoccluded upper second deciduous molar with successor	Extraction and wait and watch	Premolar spontaneous eruption

C. R.: Case Report; L. O. S.: Longitudinal observational study; F.: Female; M.: Male.

## Data Availability

Not available.

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
