# Peer review of "Therapeutic Strategies of Primary Molar Infraocclusion: A Systematic Review"

_children, 2023, doi:10.3390/children10030582_

Round 1

Reviewer 1 Report

The study is a systematic review of the Therapeutic Strategies of Primary Molar Infraocclusion. The manuscript addresses interesting aspects of the subject and has a scientific merit.

Some points could be improved to allow a better understanding.

- Did the authors not find previous reviews about this theme?

- In the results: Table 2 could also present the year of publication and the type of study (case reports, randomized trials, retrospective, and observational studies) selected.

- The conclusion should focus to answer the question raised in this study “proper therapeutic strategies of primary molar infraocclusion".

Author Response

Dear Reviewer,
Thank you for carefully reading our manuscript. We are glad you found it interesting. We thank you for giving us useful suggestions to improve our article. Below you will find our responses to your suggestions. In addition, we have made the changes on the text with
highlights.

Q: Did the authors not find previous reviews about this theme?

A: A number of reviews have been found on the same topic, however, the one presented by us encompasses a more recent period and attempts to update the guidelines already present

Q: In the results: Table 2 could also present the year of publication and the type of study (case reports, randomized trials, retrospective, and observational studies) selected.

A: The type of articles was already present in the table in the form of acronyms. The years of publication of the articles used for the review were entered

Q: The conclusion should focus to answer the question raised in this study “proper therapeutic strategies of primary molar infraocclusion".
A: Thank you for the advice, we have integrated the conclusions.

Reviewer 2 Report

Dear authors, thank you for submitting the manuscript entitled "Therapeutic Strategies of Primary Molar Infraocclusion: a Systematic Review"

Taking all these factors into the Systematic Review, if an infraoccluded or ankylosed primary tooth is observed, it should not be neglected. The ratio of children affected by infraocclusion is not negligible. Early recognition and the appropriate intervention at the right time will make for a much less complicated treatment plan with long-term results.

I have a few concerns after reading your submission:

-Please provide the criteria taken into account when considering invasive procedures.

-Describe the strategies do you have if the patient has diseases or specific clinical conditions.

-Please describe the criteria if there is radiographic diagnosis.

-In light of your statement that you do not find enough evidence to support etiology concerns, what does histology indicate.

-The manuscript will be more attractive if you display some photos or schematic drawings to illustrate the classification.

Thanks.

Author Response

Dear Reviewer,
Thank you for carefully reading our manuscript. We are glad you found it interesting. We thank you for giving us useful suggestions to improve our article. Below you will find our responses to your suggestions. In addition, we have made the changes on the text with
highlights.

1 -Please provide the criteria taken into account when considering invasive procedures.
Thank you for your kind suggestion, we have rephrased the concept on lines 371-373.

2 -Describe the strategies do you have if the patient has diseases or specific clinical conditions.
We have specified this point on lines 404-407.

3 -Please describe the criteria if there is radiographic diagnosis.

Thanks for the suggestion, the criteria for radiographic diagnosis were described in lines 77-80.

4 -In light of your statement that you do not find enough evidence to support etiology concerns, what does histology indicate

Thank you we have integrated as suggested to lines 61-70

5 -The manuscript will be more attractive if you display some photos or schematic drawings to illustrate the classification.
Thanks for the advice, we have included a graphic diagram of the classification.